# CUD-NET: Color Universal Design Neural Filter for the Color Weakness

## Abstract

Information on images should be visually understood to anyone, including the color weakness. However, it is not recognizable if color that seems distorted to the color weakness meets an adjacent object. We suggest CUD-NET[1] based on convolutional deep neural network to generate color universal design (CUD) images that satisfy both color preservation and distinguishment of color for input images. CUD-NET regresses the node point of the piecewise linear function based on information of input images and comprises a specific filter per image. We present the following methods to generate CUD images for the color weakness. First, we refine the CUD dataset on specific criteria by color experts. Second, the input image information is expanded through the pre-processing specialized on the color weakness vision. Third, we suggest a multi-modal feature fusion architecture that combines features to process expanded images. Finally, we suggest a deformable loss function by the composition of the predicted image through the model to avoid the one-to-many problems of the dataset.

## 1 Introduction

### 1.1 Motivation

The green and red color blindness are made up of 8% of males and 0.5% of females in Northern European descent[Won11], which is almost up to rate of one person in 20 people. Green and red blindness is the most common pattern, followed by blue, yellow, and total color blindness. In this paper, we generate Color Universal Design (CUD) images, which are color weakness friendly design forms, through deep learning around the aspect of the red color weakness (protanopia) and green color weakness (deuteranopia) vision. Protanopia is insensitive to red color and deuteranopia is insensitive to green color, although it varies depending on individual color weakness extent.

There are studies that help color discrimination to the color weakness, including wearable devices and surgeries[VZCR20]. However, since these research require time and cost, we simply generate CUD images with an image enhancement method based on deep learning to make the corresponding color visible for the color weakness. For an example of the left-above image $I$ in Figure 1, the people who are not color weakness can distinguish the letter '5' in the image. But as a deuteranopia vision in left-below image $I^d$, the surrounding color and the letter '5' are very analogous, making it ambiguous to distinguish the bound of adjacent object. The right-bottom target image $T^d$, refined image by color expert designers, shows that the letter '5' appeared well at the deuteranopia vision. Here, we define the non-CUD objects as the letter '5' and surroundings invisible to deuteranopia vision in the image $I$, and define the CUD objects as the letter '5' and surroundings visible to deuteranopia vision in the image $T$. In other words, CUD object means that adjacent objects are distinguishable on both the

---

[1] Code available at `https://github.com/Anonymous68864576/CUD-NET-anonymous`

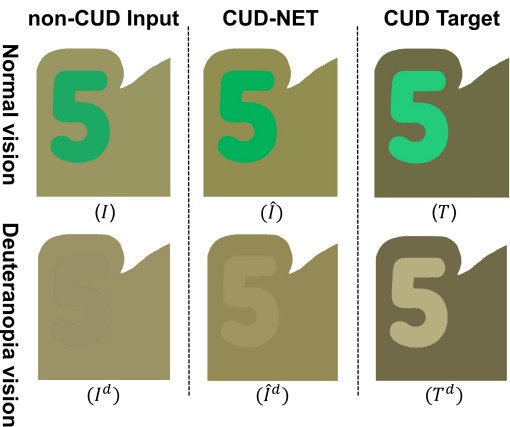

Figure 1: Comparisons of the non-CUD image, our CUD-NET's predicted image, and CUD image. The above row is represented in normal vision, and below row is represented in deuteranopia vision.

normal vision and the color weakness vision. The non-CUD object means that adjacent objects are distinguishable on normal vision but not the color weakness vision. Consequently, we generate $\hat{I}$ that satisfies CUD with a specific filter to the image $I$.

We want to apply as weak filter as possible to CUD objects to preserve color, which requires a certain level of object comprehension mechanism to do so. There are various studies from classic PCA[WEG87] to machine learning-based object segmentation methods[TSC20, ZGL*20] to define specific objects or areas in image. The research on semantic segmentation, which even provides labels between objects, seems that deep learning still does not have a complete comprehension of all objects in the real-world. The visual question answering to arbitrary questions about object's interactions, the most general issue on comprehension of object, does not have high transmission power to be practical uses[AHB*18, KZG*17, LYL*20]. Therefore, we expand feature of the input image around the information of color weakness vision and define the robust neural filter. In summary, we suggest a CUD-NET that generates an image suitable for CUD, while complying with the color preservation for the source image.

In this paper, we suggest the Color Universal Design Network (CUD-NET) to satisfy both color preservation and contrast of non-CUD objects (CUD suitability). We introduce 4 core contributions of CUD-NET.

- **Dataset refinement criteria for CUD image** We refine training data into two groups, the one with a simple color tone image based on H and V in the HSV color space, the other with two or more non-CUD objects that must be distinguished in publications.

- **Image pre-processing for CUD-NET** We carry out pre-processing to expand the information of the input image. Input image $I$ is reconstructed with three expanded feature information with noise removed.

- **Multi-modal feature fusion architecture** We define a feature layer, the fusion layer, and a regression layer to handle pre-processed images. The three features from the feature extracting layer are combined into the one fusion feature, and finally a filter is constructed by regressing the node point of the piecewise linear function, or indicator of filter.

- **Variational loss function** We suggest a deformable loss function by the composition of the predicted image through the model. Our data have a problem of one-to-many, where the specific color in input image $I$ is mapped into multiple colors in target image $T$.

## 1.2 Related Works

**Image-to-Image translation based on GAN** GAN is used in various image translation areas, including image generation, style transfer, and colorization[KWK21, IZZE17]. In a preliminary experiment, Cycle-GAN[PEZZ20] has reached the best performance in maximizing the contrast of

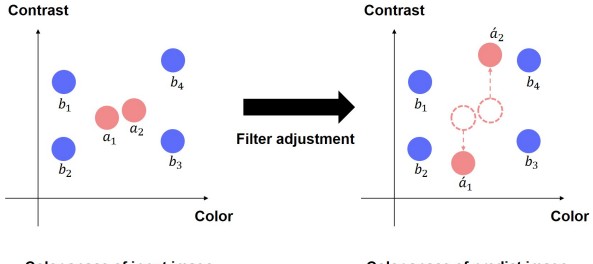

Figure 2: The ideal color conversion of predicted image between contrast and color preservation. Non-CUD object $a$ should increase the gap compared to the input image and preserve its original color, while the CUD object $b$ maintain both contrast and color.

non-CUD objects. However, our goal is to keep the color preservation of the input image as well, so in the case of black color, which has lost all its color of the input image, it is considered the worst case for color preservation. Enlighten-GAN[JGL*21] complements those instability, enabling them to generate more stable results on color preservation. But since most of the GAN-based image translation fixes the size of the predicted image, reshaping a high-resolution image causes information loss of source image. Also, it is difficult to reconstruct the complete geometry for the source image as it generates images through the dilated convolution layer.

**Image enhancement based on neural filter estimation**   Unlike GAN, there are researches that scale the pixel values of images based on neural filter estimation[WZF*19, DLT18, BCPS19]. Zero-DCE[GLG*20] is a low-light image enhancement research that provides a brighter visual display of input image. It estimates pixel-wise and high-order filter for dynamic range adjustment of input images with lightweight deep network, DCE-Net. DeepLPF[MMM*20] tried to solve the problem by using a graduated filter, elliptical filter, and polynomial filter. The authors not only tried to visually enhance the contrast of images but also to comprise stable filters that are easy to understand for the spectators while keeping the color preservation. In our problem, however, the contrast factor is almost same results as the input image in both visions, while complying the high color preservation, resulting over-stable filter. It is assumed that the inability in comprehension of object's interaction leads to over-stable filter.

## 2   Methodology

We define the ideal predicted image as an increase in the contrast between non-CUD objects and the color preservation for the input image. The non-CUD object $a$ should be mapped into $á$ and CUD object $b$ should preserve its color and contrast like an ideal example of Figure 2. However, as our neural filter affects the whole pixels throughout the image, we have the constraint of applying the same filter to objects $a$ and $b$. It is very hard to make the contrast and color of object $b$ exactly the same as before the filter adjustment while maximizing the contrast of object $a$. Therefore, we propose a deep learning-based regression to comprise the specific filter per image that maximizes the contrast of object $a$ while minimizing the adjustment of features on object $b$.

First, we propose a solution to maximize the contrast of the L channel values in CIELab color space[RG19]. We empirically confirmed that protanopia and deuteranopia, which account for the most proportion of color weakness, can distinguish the difference by L channel values in common when the non-CUD objects are adjacent to each other. To illustrate Figure 1 again, the L channel value of letter '5' in image $I$ is 61 and the surrounding color is 61. The distinguishment between the two objects is easy to normal vision, however the image $I^d$, the deuteranopia vision, is very ambiguous. On the contrary, the CUD target image $T$ and $T^d$ have a difference of L channel value 75 for the letter '5' and 45 for the surroundings, making it easy to distinguish between the normal and the deuteranopia vision. Due to the characteristics of these data, we refine a data pair by defining a criterion that separates two invisible non-CUD objects by L channel values.

Secondly, we propose a variational loss function and multi-modal feature fusion network for color preservation. It can be said that the increase in the contrast of L channel values between non-CUD

objects is quantitatively superior, but not in the case of increasing the differences in color preservation of input images. When non-CUD objects exist, as a simple example, the most likely way to maximize contrast is to polarize the color of the object black and white. But it is the result of complete ignorance for color preservation, so just enabling to distinguish between non-CUD objects is not always a good answer. A strong filter must be applied to distinguish non-CUD object, but its impact should not be too extensive to leading the loss of information in CUD objects. In this paper, we solve this problem by taking an appropriate trade-off of color preservation and contrast of L channel value.

## 2.1 Dataset refinement criteria for CUD image

The training data is refined by two groups. The one is vectorized image with two colors divided by value V and hue degree H in HSV color space[HMKO19], and the other is image with two or more objects that must be distinguished while preserving the color of non-CUD objects. The training data is grouped about 1,600 color combinations into the same V and then simulates them with the deuteranopia vision, converting to the adjacent color family to comply color preservation. All conversions are scaled within only S and V in HSV color space to increase at least 15 difference in the L channel value of selected non-CUD objects. The colors are combined with the 10 essential H and tones, and the similar color simulated with the deuteranopia vision was converted. Consequently, the key part of refining training data is preservation of color, allowing the models to comply with the same approach on learning.

## 2.2 Image pre-processing

Our model regresses node points of piecewise linear function, which will be described in the model architecture section, and the final filter is a multiplication operation for the input image. Therefore, the multiplication operations of less than the number 1 tend to fade the color saturation. The image without color inversion converges the white color value to 1, so if the multiplying value is in the [0, 1] range, the white color is shifted to the black. By inverting the color of input image, it ignores the multiplication operations for white value with 0.

We generate the map image $I^m$ based on original RGB input images calculating the difference value between the image with an aspect of normal vision and the image with an aspect of deuteranopia vision. Recent studies have been conducted to augment the information or expanded the models' perspective through transformer models[JSZK16, RFB15]. In our experiment, however, the transformer model tends to generate the predicted image ignoring the source color, which result in the polarized color to black and white like Cycle-GAN's.

$$I^m = \left| invert(I^n) - invert(I^d) \right| \quad (1)$$

$$I = \delta \left( cat^{channel} \left( I^n, I^d, I^m \right) \right) \quad (2)$$

After applying color inversion from the original RGB input image $I^n$, we generate the image $I^d$ with an aspect of deuteranopia vision in equation 1. From these two generated images, we can get the absolute difference value to compose the map image $I^m$. In equation 2, the final input $I$ concatenated with $9 \times H \times W$ dimensions passes through the model. The $\delta(.)$ clips output to a range of [0, 1].

## 2.3 Model Architecture

CUD-NET regresses the node points of piecewise linear filter function from the input. The value of each node points computes the multiplication operation and generates the predicted image. The input $I$ is compressed into 3 feature blocks matching each input through convolution layer, pooling layer, and global pooling layer. The input with 9 channels is separated into $3 \times 3$ channels before passing the model. The first 3 channels are literally used as the main inputs, where the multiplication operation takes place, while the remaining 6 channels are used as features.

First of all, we use multi-modal fusion architecture for three separate inputs to extract expanded features. The three inputs converted to the HSV color space pass through a weights-sharing convolution layer to extract a feature block corresponding to the inputs. Each convolution layer consists of kernel size=3, stride=1, and padding=1, reducing dimension through average pooling. We empirically

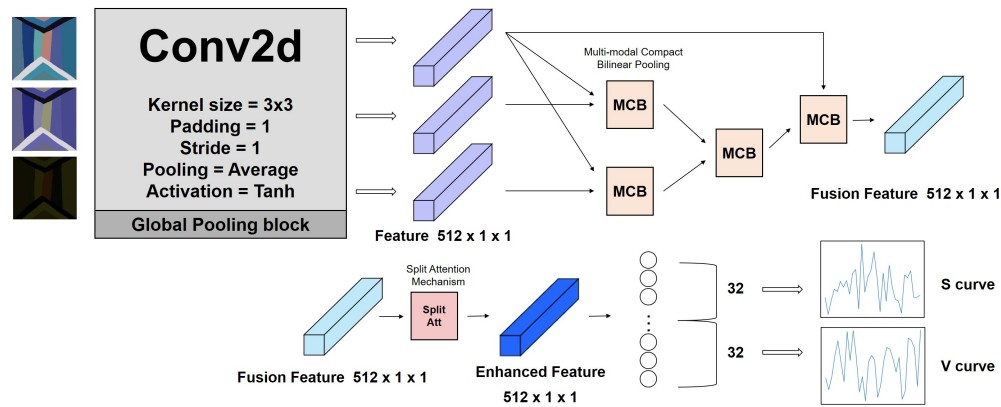

Figure 3: Overview structure of CUD image generation

noticed that the most values of output feature have distribution within the range of [-1, 1] with valid values for constructing the node points, so we use hyperbolic tan for activation function. Since we use inputs with unstructured image size, the last global pooling block holds the size of the feature instead of the average pooling block[LCY14].

The three feature blocks are combined through the multi-modal compact bilinear pooling gate (MCB)[FPY∗16], following the fusion process shown in Figure 3. The MCB gate allows both features to interact in a multiplicative way with low memory consumption and computation times. The fusion features are complemented to enhanced feature through the split attention mechanism[ZWZ∗20]. At the beginning of the experiment, we have applied the convolutional block attention mechanism[WPLK18] of each MCB gate, but we found that it does not make sense of understanding the feature itself, so we apply only one attention block to the last fusion feature.

The enhanced feature pass through the fully-connected regression layer. We picked the 64 points to be regressed to compose the piecewise linear function, which is empirically confirmed to the optimized number of points in this research. The first half of the values construct the node points of the S channel and the other half comprise the V channel in HSV color space. Finally, node points become the scaling factors to generate predicted image in equation 3[MMS19].

$$S\left(I_i^{s,v}\right) = k_0 + \sum_{m=0}^{M-1} \left(k_{m+1} - k_m\right) \delta\left(MI_i^{s,v} - m\right) \quad (3)$$

The total number of node point $M$, each pixel values of S, V channel in input image $I_i^{s,v}$ are multiplicated with the slope of actual regressed value $k_m$, the $m-th$ generated node point. The specific node points $M$ is scaled through a multiplication operation to pixel value of the input image according to each node point.

## 2.4 Loss function

Our dataset has one-to-many problems between input and target data. In dataset pair $(I_1, T_1)$, $(I_2, T_2)$, ..., $(I_n, T_n)$, for example, the red color in $I_1$ can be targeted to purple color in $T_1$, and the red color in $I_2$ can be targeted to orange color in $T_2$. With these one-to-many dataset structures, we design the loss function $\mathcal{L}$ that expresses the potential and the diversity of predicted image in equation 4.

$$\mathcal{L} = \sum_{i=1}^{N} Lab_{loss}\left(V\left(\Phi\left(\hat{I}_i\right)\right)\right) + H_{loss}\left(\Phi\left(\hat{I}_i\right)\right) \quad (4)$$

**Stencil Masking** As explained in the dataset refining criteria, we do not proceed with color conversion for all areas in the target images, but only for areas with color combinations that are

invisible to the deuteranopia (non-CUD object). For this reason, the input image has color regions of converting color and unconverting color, which also can be referred to as non-CUD object and CUD object. To imply the color bound to model, the stencil masking method is introduced.

$$\Phi\left(\hat{I}_i\right) \;=\; \hat{I}_{ij} \;||\; \left(I_{ij} \cdot T_{ij}\right) \quad (5)$$

We consist a stencil maps through the logical and operations '·' of each pixel value $I_{ij}, T_{ij}$. Stencil map can specify the non-CUD area and be computed with predicted image $\hat{I}_{ij}$ of logical or operation '||'in equation 5. Consequently, CUD object of the image adjusted with a stencil mask does not carry out the neural filter computation, such as the same way we refine the target image. This refined image is calculated on the loss function.

**CIELab Loss** We use the CIELab channel loss function to maximize the contrast of color on deuteranopia vision. To stabilize the contrast and brightness of the predicted image, we calculate the MS· SSIM(multi-scale structural similarity[WSB03]) of L channel.

$$Lab_{loss} \;=\; \left\| Lab\left(\hat{I}_i^{rgb}\right) - Lab\left(T_i^{rgb}\right) \right\|_1 \;+\; MS\!\cdot\!SSIM\left(Lab\left(\hat{I}_i^{L}\right),\; Lab\left(T_i^{L}\right)\right) \quad (6)$$

The $Lab\left(.\right)$ expression in equation 6 returns the CIELab channel corresponding to the RGB channel, and all calculations are made only on the L channel.

**Histogram Loss** We use the histogram loss function to comply with the color preservation of the image. The RGB channel is used to preserve its color, contrary to using only the L channel in other loss functions. Handling the RGB channel as a loss function rather than using Lab's ab channels has shown better results on color preservation.

$$H_{loss} = -\omega_{hist} \int N\left(\hat{I}_i^{rgb};\, \sigma\right) - N\left(T_i^{rgb};\, \sigma\right) \quad (7)$$

When simply designing a loss function with the L1 distance of the RGB channel pixel values, it was very sensitive to certain values and the gradients are diverged, resulting in an untrainable experiment. Therefore, we used a gaussian expansion method[SAC*17] denoted by $N(.)$ to infer a differentiable histogram loss function in equation 7. We compute the difference of the RGB channel of the differentiable histogram function, which can be altered to mean squared error or cosine similarity. The scaler $\omega_{hist}$ is determined in inverse proportion to the size of the input image. By maintaining the RGB similarity between the predicted image and the target image, we can comply with the color preservation.

**Variational Prediction** There are various ways to maximize difference of the L channel in the image. And the target image is converted at least two colors compared to the input image. However, the predicted image of the model is generated by the neural filter, so it is unpredictable which area of color is modified. Therefore, if the color in predicted image is over-shifted or in the color value of opposite shifts to the target, the loss will rather increase. In addition to one-to-many problem that the data pair itself does not matches one-to-one in a particular color, it is necessary to generate alternative predicted image with the same aspect of the data pair. We calculate the loss function with a variational prediction based on the predicted image for potential color shifts.

The first potential is the case of excessive shifts. Assume that $I_i^L = \{74, 41, 79\}$, $\hat{I}_i^L \;=\; \{97,\, 10,\, 70\}$, $T_i^L \;=\; \{50,\, 41,\, 80\}$ in L channel value. The first and third components of each image is non-CUD objects, and second component is CUD object. Therefore, we refined data paired with a difference of 15 on L channel. Here, we clip the excessive L channel value in $\hat{I}_i^L$ by equation 8. Up to this point, no calculation is made as no value is exceeded in this example. The second potential is the case of opposite shifts. It can be said that a complete neural filter has been proceeded for value 97, 10, 70 where L channel difference is 27. However, if we actually calculate the mean square error between $\hat{I}_i^L$ and $T_i^L$, it will be an large value over 1k. Here we can generate alternative predicted image from equation 9 and 10.

$$clip\left(\hat{I}_{ij}\right) = \begin{cases} max\left(\hat{I}_{ij},\ T_{ij}\right), & I_{ij} > T_{ij} \\ min\left(\hat{I}_{ij},\ T_{ij}\right), & I_{ij} \leq T_{ij} \end{cases} \quad (8)$$

$$R_1 = 2I_{ij} - \hat{I}_{ij}, R_2 = \hat{I}_{ij} \quad (9)$$

$$V\left(\hat{I}_{ij}\right) = argmin\left(\|clip\left(R_{1,2}\right) - T_{ij}\|_2\right) \quad (10)$$

As mentioned above, we define thresholds by the maximum and minimum value of each corresponding pixel position of $\hat{I}_i^L$ and $T_i^L$. By computing a difference of residual map and the input image, we induce the alternative two images $R_1$, $R_2$. As a result, $\hat{I}_i^L$ with a smaller L2 distance is selected to alternative predicted image in equation 10, and it is finally computed with loss function compared to the $T_i^L$. The above equation establishes $V\left(\Phi\left(\hat{I}_i\right)\right)$ = {54, 41, 80} and the mean square error to the target image is approximately 5, which is agreeable loss value respect to $\hat{I}_i^L$ itself.

**Identity Loss[ZPIE17, TPW16]** We use $\mathcal{L}_{identity}\left(T_i\right)$ to apprehend the CUD object to the model. In the case of target image that already satisfy the CUD, the filter should be relatively weakly applied than input image. The input of identity loss is target image $T_{ij}$ instead of input image $I_{ij}$, and the reference of the loss function is also target image $T_{ij}$ to maintain the value itself. In computing identity loss, we do not require variational prediction as we cannot judge the potential region by equation 10.

# 3   Experiments

The experiment was performed with Tesla V100 SXM2 and Intel Xeon Gold 5120 and the computation speed was about 40 images per minutes. We refined a dataset with Adobe Photoshop to maximize contrast in the L channel by adjusting saturation and brightness for areas that require color conversion based on deuteranopia vision simulation. Color experts has refined about 1,500 vectorized image for the training data and 300 publication images for the validation data. All the comparative experimental models used the same train, test, validation data in this paper. We used the inference data in publications, which is almost composed of vectorized images, as colors often appear distorted in a gradation-rich image. The Figure 4 is arranged in descending order of the number of combinations in colors from the top image.

Both structure similarity (SSIM)[ZBSS04] and peak signal to noise ratio (PSNR) in Table 1 can indicate whether the image is suitable for CUD or not. As a notable aspect, the result has shown that comparative models with lower metrics are sensitive to high-gradation input images, which generated color-heterogeneous image. SSIM and PSNR itself can determine the increase in contrast compared to the target image but do not determine whether the color preservation complied. Therefore, we evaluated SSIM and PSNR with three references, inputs images $I$, predicted images $\hat{I}$, and target images $T$. The higher the estimation of the $\hat{I}$ and $I$, the more color preservation factor worked. The higher the estimation of the $\hat{I}$ and $T$, the more increase in contrast can be considered. We also define SSIM mean absolute error, PSNR mean absolute error to measure the extent of the conversion between the $F : I \rightarrow T$ and $F : I \rightarrow \hat{I}$ in equation 11 and 12, respectively. The $N$ is total number of inference data.

$$SSIM{\cdot}MAE = \frac{1}{N}\sum_{i=1}^{N}\left| SSIM\left(\hat{I}_i,\ T_i\right) - SSIM\left(I_i, T_i\right)\right| \quad (11)$$

$$PSNR{\cdot}MAE = \frac{1}{N}\sum_{i=1}^{N}\left| PSNR\left(\hat{I}_i,\ T_i\right) - PSNR\left(I_i, T_i\right)\right| \quad (12)$$

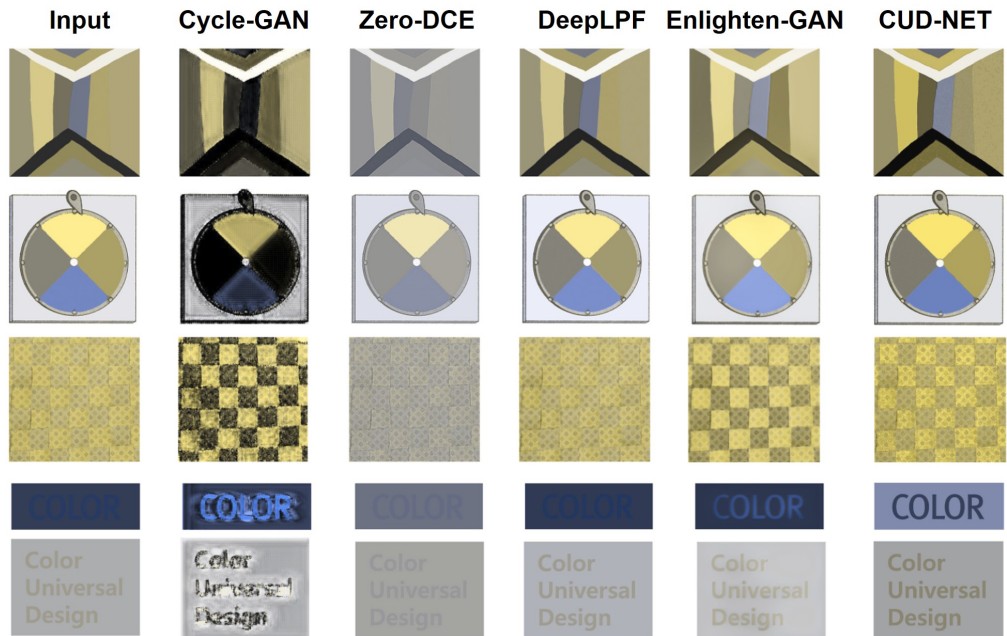

Figure 4: Comparisons of predicted images in deuteranopia vision. The color experts selected the validation data that do not satisfy the CUD in publications.

| Architecture | SSIM$\left(\hat{I}, I\right)$ | SSIM$\left(\hat{I}, T\right)$ | PSNR$\left(\hat{I}, I\right)$ | PSNR$\left(\hat{I}, T\right)$ | SSIM·MAE | PSNR·MAE |
|---|---|---|---|---|---|---|
| Cycle-GAN | 0.630 | 0.634 | 13.28 | 13.83 | 0.3191 | 8.4430 |
| Zero-DCE | 0.924 | 0.888 | 21.95 | 18.67 | 0.0661 | 3.7300 |
| DeepLPF | 0.850 | 0.831 | 26.31 | 20.34 | 0.1220 | 2.0566 |
| Enlighten-GAN | 0.820 | 0.808 | 21.85 | 19.58 | 0.1470 | 3.9983 |
| Enlighten-GAN(scaled) | **0.966** | 0.921 | 24.86 | **21.36** | 0.0392 | 3.4937 |
| CUD-NET(low bottle-neck feature) | 0.897 | 0.866 | 27.77 | 21.01 | 0.0901 | 2.0826 |
| CUD-NET | 0.962 | **0.924** | **29.54** | 21.19 | **0.0312** | **1.4760** |

Table 1: Evaluation table of comparison experiment. The CUD-NET with a low bottle-neck feature achieves better results in the experiment of the deuteranopia and the protanopia subjects(Figure 5), although the evaluation metrics are lower than that of CUD-NET.

Cycle-GAN and Zero-DCE showed worse result than others. Cycle-GAN model had difficulty reconstructing a geometry of a particular object, and overall color had low saturation and brightness, resulting in color conversion into an almost grey scale image. Zero-DCE is faded in color, and the contrast was not much different from the input image. The overall image lost its color preservation, which we focused to solve in this paper.

DeepLPF meets both the color preservation and contrast that we deal with for. However, DeepLPF tends to color be over-stably filtered for the images with fewer color combinations. Although the color preservation has complied better than other experiments, there were many failed results from the perspective of contrast, which the over-stable filter leads to by DeepLPF.

Remarkably, predicted images of Enlighten-GAN showed reasonable results. However, simple color combinations or the images with already satisfying the CUD often showed results degenerated with low CUD suitability. Enlighten-GAN was able to generate the results we targeted, but its deviation of filter is so high that it sometimes failed to satisfy the contrast even on simple images or decreased the contrast. As the problem of GAN-based method including Enlighten-GAN, moreover, model fixes the width and height of the predicted image. If width and height of $T$ and $I$ down-scaled

to size of Enlighten-GAN $\hat{I}$(approximately 25K pixels in this experiment), the $SSIM\left(\hat{I},\ I\right)$ and $PSNR\left(\hat{I},\ T\right)$ showed higher estimation in some metrics than CUD-NET. In the opposite case of $\hat{I}$ up-scaled to size of $T$ and $I$, the significantly low estimation was recorded due to the information loss of up-scaling problem.

CUD-NET showed stable and robust predicted images in both color preservation and increase in the contrast compared to other experiments. In comparing the values in the same region of $I$ and $\hat{I}$, the model scaled two L channel values with opposite side in the most of case, the one goes up and the other goes down. When we reduced the number of bottle-neck feature of model, it tends to record relatively high deviation of filter scales according to the number of combinations of colors. In summary, the CUD-NET showed the highest estimations for 4 evaluation metrics. Moreover, as our model adopted a neural filter unlike generation models, there is no loss of information regarding the scaling of predicted images.

Figure 5: The box bar is ordered to the left side, input image $I$, Enlighten-GAN, DeepLPF, CUD-NET. The y position of box bar represents a mean and length of the box bar represents a deviation of each experiment. The lower the graph is, the higher the rank is.

The figure 5 shows the evaluation of the deuteranopia and the protanopia. The evaluation metrics consist of object distinguishability and color harmony in order of input image I, predicted image of Enlighten GAN, DeepLPF, and CUD-NET. User study has tested upon the total of 6 subjects, 4 deuteranomaly and 2 protanomaly. The subjects were asked to list the ranks of object distinguishability and color harmony of 4-paired-image for each model-blinded item. As the experimental results, the deuteranopia subject ranked the 1-st in the object distinguishability of CUD-NET at an average rank of 1.821, followed by Enlighten-GAN at an average rank of 2.512. Similarly, the protanopia subject also ranked the 1-st in CUD, followed by Enlighten-GAN, DeepLPF, and input images. The evaluation of color harmony showed that the subjects tend to assume that the image with a good object distinguishability has good color harmony preferentially. For a total of six subjects, the five subjects chose the CUD-NET, with the exception of one who ranked Enlighten-GAN by a subtle gap

## 4 Conclusion

In this paper, we proposed deep network to generate CUD images from non-CUD input images. The pre-processing and multi-modal fusion layer could comprehend the information for color weakness, and the variational loss function makes the model further adapt to CUD dataset. Compared to other research, we are able to maintain high-resolution images and both stable color preservation and contrast with neural filter per images.

Our current research shows a robust filter for a single color, such as vectorized images, but it is difficult to expect stable results in the case of a real-world image with high gradation in hues. We consider the same limitation of our work when the certain pixel values react sensitively, making noise appear more prominent in the predicted image. In the future, we plan to create additional datasets with gradation on the vectorized image and focus on the fusion layer to improve performance of the model.

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
