# OpenReview forum: "CUD-NET: Color Universal Design Neural Filter for the Color Weakness"
_NeurIPS.cc/2021/Conference — NeurIPS 2021 Submitted_

### Official Review · Reviewer_r6iV · 2021-07-16

**Rating:** 3
**Confidence:** 5

**Summary:**

The paper proposes a deep-learning based method to create (vector-graphic) images that enables color deficient viewers to identify every object in the scene. The method boosts the contrast and/or modifies the hue of colors that are confusing to protanopes and deuteranopes. The method has a pre-processing step, a multi-model feature fusion architecture, and a variational loss function. Comparisons were carried out with GANs and image enhancement methods.

**Limitations And Societal Impact:**

The limitations and societal impact is mentioned

**Main Review:**

While the idea and project is certainly worthwhile pursuing as it benefits vision impaired people, this paper is not yet ready for publication due to several issues addressed below:

-	SOTA: There are a large number of algorithms out there that correct images for protanopes and deuteranopes, many of them listed in reference [RG19]. A discussion of some of them, conceptually closest to the method presented here, would improve the SOTA. But more importantly, a comparison to some of these methods in the Experimental Section would very much strengthen this paper.
-	Experiments: As already alluded above, I do not consider the comparison to GANs nor image enhancement methods sufficient to validate the proposed methodology. A comparison to at least some of the color-blind correction methods is needed. Also, PSNR and SSIM are not really relevant metrics here, as they do not measure perceptual color differences. Some Delta E variant would be more appropriate. I appreciated that a psychovisual study was done, but that one needs to be better described on what exactly the methodology was.
-	The proposed method so far only works for vector graphics. There are of course some applications for that, but it is limited. I encourage the authors to also extend the method to natural images. I do acknowledge that creating an appropriate ground truth database will not be trivial, thus I find a comparison to existing color deficiency correction algorithms even more important.
-	Lab Loss Function: I do not understand the motivation for the Lab loss function. Perceptual differences in Lab Color space are measured as an L2 distance which is equal to Delta E, an (almost) perceptual metric. Here L1 is used, as well as MS SSIM. Considering we are dealing with vector graphic images I am not sure SSIM is meaningful. I would like to see a better discussion of this Loss function.
-	English: The paper is hard to read as there are many grammatical mistakes. A good proof-reading by a native English speaker will certainly improve the quality of the manuscript.


**Time Spent Reviewing:**

2h

---

> ### Author Response · Authors · 2021-08-19
> **Review Feedback**
>
> Thanks for your sincere review.
>
> SOTA
>
> -> Our research is focused on the fusion of machine learning and color weak deficiency. The classic algorithm is not robust to image complexity, and we think machine learning would understand the image complexity as the robust filter is made up. The writing process is now in progress for listed references[RG19].
>
> References and metrics.
>
> ->The reference baselines are adapted through both targets, the color naturalness and image enhancement under the machine learning. As the image enhancement research on the color universal design is not proceeded before (in our searches), the comparison in related works could be not proper as you insist. We agree it is hard to judge that our research is in the middle of image enhancement and the recoloring algorithm field. So the MS SSIM and PSNR is adapted to evaluation metrics, which is the most common metrics in image enhancement filed, and also we are very thankful your opinion using delta e perceptual metric.
>
> Limitations
>
> -> As our initial goal is set to the publications, our first step is ensuring of our research working. Our next goal is extending the method to nature images as the dataset is produced.
>
> English
>
> ->The paper writing and grammatical error is now on correction by English native.

---

### Official Review · Reviewer_Sm8T · 2021-07-17

**Rating:** 4
**Confidence:** 4

**Summary:**

This paper focuses on color universal design to satisfy both color preservation and distinguishment of color for the color weakness. To achieve that, the paper refines the color universal design dataset on specific criteria by color experts, expands the input image information via pre-processing, proposes a multi-modality feature fusion architecture in CNN, and suggests a deformable loss function. The experiments are conducted on the provided dataset. Besides, a user study is carried out.

**Limitations And Societal Impact:**

The author has clearly presented and solved the limitation and potential negative societal impact.

**Main Review:**

Pros:

+This topic is interesting and meaningful. Although there are numerous image enhancement methods, they are not suitable for color universal design. The paper raises this topic and provides some insights.

+The provided dataset and pre-processing are valuable for this task.

+The paper provides sufficient experimental results in the supplementary material.

Cons:

-Although a new research topic is raised in this paper, the novelty of the paper is limited to me. Besides the dataset, the multi-modal feature fusion, and loss function are intuitive and straightforward. What the results would be like if just using a simple network such as a Unet-like network? With the same training data, a simple network may achieve similar performance under the constraint of the l1 or l2 norm.

-The experiment is not much convincing to me. The proposed method is compared with several image enhancement methods. It is not clear if the compared methods are training using the same training data as the proposed method. Some compared methods are un-supervised or zero-reference-based. Thus, strong baselines should be included for comparison. For example, a supervised method is trained under the same training data as the proposed method. The subjects in the user study are only six. More subjects should be invited for a stable statistic. Moreover, the performance of the user study shows that the proposed method just achieves relatively good performance when compared with these baselines. However, the accuracy of the proposed method is far from the practical applications in real cases.


**Time Spent Reviewing:**

10

---

> ### Author Response · Authors · 2021-08-19
> **Review Feedback**
>
> Thanks for your sincere review.
>
> Model Architecture
>
> ->The U-Net-like network is performed in the initial baseline research, but it is failed with gradient vanishing problem and weak filter composition. We thought that the model could had not understand the correlation between input image and target image, so we increase the information of input by deuteranopia vision(Daltonization), not neural networks structure([MMS19] also insist similar argument).
>
> Experiments
>
> ->All of the referenced baselines are first inferenced with our target dataset. Its results is good enough for the nature image, but it frequently failed on the very simple image(with only 2 tones) that should be applied strong color correction. So we trained baselines with our CUD dataset to experimental comparison, as our goal is first targeted to vectorized image. To secure more test subjects, we are now developing web-based prototype version of our researches. And we agree that it is hard to judge what the proper metric or evaluation is as our research is in the middle of image enhancement and the recoloring algorithm field.

---

> > ### Comment · Reviewer_Sm8T · 2021-08-28
> > **Response to authors feedback**
> >
> > Thanks for the authors' feedback. After reading the authors' feedback, I understand that, as a new research topic, some baselines and evaluation metrics are missing. However, it is important to convince the readers and reviewers the proposed method works for real cases. From the current results and explanations, I still concern about the applications of the proposed method in real cases and natural images.  But, this research topic is meaningful and important. From my side, I like this topic.

---

### Official Review · Reviewer_Zn84 · 2021-07-23

**Rating:** 3
**Confidence:** 3

**Summary:**

The paper proposes a method to generate CUD (color universal design) images with a deep learning based image enhancement method. The output CUD images are more discernible to those challenged by red/green color blindness (protanopia/deuteranopia). The network architecture proposed to generate CUD images consists of three steps -- i) feature layer wherein 512 dimensional features are extracted from concatenated input images, ii) fusion layer to combine the 3 features, iii) Finally an attention layer to generate an enhanced feature. Subsequently, 64 values are regressed from the enhanced feature to estimate a piecewise linear function for S/V channels of the output image. As stated in Sec 2.4, different loss functions are used -- stencil masking (only change areas that challenge deuteranopia vision), CIELab loss to enhance contrast, histogram loss to preserve original colors etc. Quantitative evaluation is performed in Tab. 1 which measures color preservation and contrast enhancement by comparing SSIM/PSNR metrics between predicted image (I^\hat) and original image (I), target image (T). In Fig. 5, protanopia/deuteranopia rank different methods based on distinguishability of output CUD image.

**Limitations And Societal Impact:**

The authors propose a method to generate images which are distinguishable to those with red/green color blindness. They perform a user study with 6 subjects to demonstrate increased distinguishability of the images generated by the proposed method. Larger study needs to be performed to ascertain the performance and failure modes of the proposed method before it can be deployed to assist those with color blindness.

**Main Review:**

Strengths:
+ The paper addresses an important problem of using deep-learning based image enhancement techniques to generate CUD images which are distinguishable by those with red/green color blindness
+ Test subjects in user study of Fig. 5 prefer the CUD images generated by the proposed CUD-NET method over other baselines and original input

Weakness:
- The details in the paper are difficult to follow and the paper writing can be improved. For example,
i) L118-119 "The training data is grouped about 1,600 color combinations into the same V and then simulates them with the deuteranopia vision, converting to the adjacent color family to comply color preservation." What is the process of simulating the images with deuteranopia vision? How does the conversion to adjacent color family work? The authors should explain these details with equations if necessary.
ii) L139-140 "After applying color inversion from the original RGB input image In, we generate the image Id with
an aspect of deuteranopia vision in equation 1." It is not clear how I^d is generated, can the authors explain?
iii) L240-242 "Color experts has refined about 1,500 vectorized image for the training data and 300 publication images for the validation data. All the comparative experimental models used the same train, test, validation data in this paper." The authors should describe how they sourced the images, what annotation/enhancement process was used to generate supervision (i.e. enhanced/CUD images), and what are the train/val/test splits?

- Sec 2.4 describes various loss functions, however it is unclear how each of them helps improve the performance. An ablation study should be performed by training the network with different loss terms and measuring the effect on quantitative/qualitative metrics in Sec 3.

- Sec 2.3 describes the model architecture, however some design choices are not explained clearly
i) For shared convolutional layers, why is a standard architecture (possibly with pre-training) e.g. residual network not used?
ii)  Does pooling and split attention mechanism help improve the performance? Can the authors demonstrate that using pooling + attention is better than simply concatenating the 3 512-dimensional features and regressing the 64 values of piecewise linear function?

The above weaknesses need to be address for the reader to get insights into the proposed method and its results.

**Time Spent Reviewing:**

3

---

> ### Author Response · Authors · 2021-08-19
> **Review Feedback**
>
> Thanks for your sincere review.
>
> The paper writing and grammatical error is now on correction by English native.
>
> "The training data is grouped about 1,600 color combinations into the same V and then simulates them with the deuteranopia vision, converting to the adjacent color family to comply color preservation"
>
> -> The Daltonization is used for deuteranopia vision conversion, and we will add its reference on the paper.
>
> "After applying color inversion from the original RGB input image In, we generate the image Id with an aspect of deuteranopia vision in equation 1"
>
> -> The color inversion simply means the reverse of each channel in color space. The image Id is generated from above method, Daltonization.
>
> "Color experts has refined about 1,500 vectorized image for the training data and 300 publication images for the validation data. All the comparative experimental models used the same train, test, validation data in this paper."
>
> -> The train data set is produced by color experts(2nd author), using S, V channel control on Photoshop and Illustrator environment. The train data is now exclusive to the partner institute, but the validation data for the publications can be found in github link, https://github.com/Anonymous68864576/CUD-NET-anonymous/tree/main/dataset/inference.
>
> The model architecture change and data set offer is proceed simultaneously, so it is not clear which components effected to the model's performance. But as the architecture changes, the model tends to preserve the original color of input image, not the color contrast, which means a weak filter applied to inputs, and the gradient sometimes vanishes as the trains going on. In that time, model apply the same filter for all datasets. So we finally consider to adapt MCB layer for feature lives until the bottle-neck layer. As you mentioned, the residual layer would help the gradient vanishing problem, but we didn’t come up with while research is going on as it comes to me not a light issue.

---

### Decision · Program_Chairs · 2021-09-27

**Decision:**

Reject

**Comment:**

Reviewers agree there are fundamental deficiencies in this submission. One is the lack of baselines for the task, some of which pre-date deep learning methods. Their addition is key to improving the paper and the comparison of the results.

All reviewers also mention the lack of clarity in the writing that makes it difficult to extract the details of the contribution. Unfortunately, I recommend rejection.